# Monitoring and Control Framework for IoT, Implemented for Smart Agriculture

**DOI:** 10.3390/s23052714

**Published:** 2023-03-01

**Authors:** Elisha Elikem Kofi Senoo, Ebenezer Akansah, Israel Mendonça, Masayoshi Aritsugi

**Affiliations:** 1Graduate School of Science and Technology, Kumamoto University, Kumamoto 860-8555, Japan; 2Faculty of Advanced Science and Technology, Kumamoto University, Kumamoto 860-8555, Japan

**Keywords:** Internet of Things (IoT), open-source, IoT architecture, smart agriculture, monitoring, control, framework, domain-agnostic, sensors, actuators

## Abstract

To mitigate the effects of the lack of IoT standardization, including scalability, reusability, and interoperability, we propose a domain-agnostic monitoring and control framework (MCF) for the design and implementation of Internet of Things (IoT) systems. We created building blocks for the layers of the five-layer IoT architecture and built the MCF’s subsystems (monitoring subsystem, control subsystem, and computing subsystem). We demonstrated the utilization of MCF in a real-world use-case in smart agriculture, using off-the-shelf sensors and actuators and an open-source code. As a user guide, we discuss the necessary considerations for each subsystem and evaluate our framework in terms of its scalability, reusability, and interoperability (issues that are often overlooked during development). Aside from the freedom to choose the hardware used to build complete open-source IoT solutions, the MCF use-case was less expensive, as revealed by a cost analysis that compared the cost of implementing the system using the MCF to obtain commercial solutions. Our MCF is shown to cost up to 20 times less than normal solutions, while serving its purpose. We believe that the MCF eliminated the domain restriction found in many IoT frameworks and serves as a first step toward IoT standardization. Our framework was shown to be stable in real-world applications, with the code not incurring a significant increase in power utilization, and could be operated using common rechargeable batteries and a solar panel. In fact, our code consumed so little power that the usual amount of energy was two times higher than what is necessary to keep the batteries full. We also show that the data provided by our framework are reliable through the use of multiple different sensors operating in parallel and sending similar data at a stable rate, without significant differences between the readings. Lastly, the elements of our framework can exchange data in a stable way with very few package losses, being able to read over 1.5 million data points in the course of three months.

## 1. Introduction

The Internet of Things (IoT) has the potential to transform a wide range of industries, including agriculture, healthcare, and transportation. It is said that it will transform different fields such as healthcare [1,2,3], education [4], transportation [5], security [6,7], finance [8,9], agriculture [10,11], and manufacturing [12]. It provides an opportunity for human-to-machine and machine-to-machine interaction [13,14] and is expected to become more common in modern society as the technology continues to be adopted more and more, and it is expected to reach about 75 billion devices by 2025 [15].

A typical IoT system describes a network of sensors and actuators that are either directly connected to cloud services or to edge devices that may be connected to the cloud [16,17,18]. The sensors perform monitoring by collecting data related to phenomena of interest, and actuators execute the control function, causing changes in the controlled devices. Different domains require different types of sensors and other devices to collect data on various aspects of the environment or system being monitored, such as soil moisture in agriculture, patient vital signs in healthcare, or vehicle localization in transportation [19,20,21]. These data are then analyzed and used to optimize various activities or processes, such as irrigation in agriculture, patient care in healthcare, or fleet management in transportation [20,21,22].

The organization of devices such as sensors, actuators, edge devices, cloud services, protocols, and layers that constitute IoT networking systems are referred to as the architecture [23]. IoT architecture is crucial to delivering desired services. Researchers have suggested multiple IoT architectures in response to a number of challenges, including interoperability, security and privacy, dependability, energy limitations, scalability, and the lack of universal standards [23]. According to Gupta et al. [24], the four major elements of the IoT environment are physical devices, interconnectivity, real-time applications, and operating platforms. These elements of the IoT environment have seen major advancements in technologies, protocols, and standards. Xu et al. [25] state that the success of IoT can be attributed to the advancements being made in many related technologies, including communication efficiency and the energy efficiency of devices, which have improved over the years. Gupta et al. [24] also state that, although these technologies, protocols, and standards are the main forces behind the expansion of the Internet of Things, they present new challenges to its integration with the traditional Internet framework and network scaling, in addition to problems unique to IoT, such as device heterogeneity and ambiguities in its framework standardization.

The adoption of full-stack development solutions for the Internet of Things is still in its early stages; thus, there are challenges that must be overcome. As expected of any technology, the increase in interest, and its adoption without widely accepted standards, has led to a fragmented ecosystem, which makes the interoperability of systems challenging. The increasing participation of people and institutions has contributed to the fast growth in IoT, and the diversity of the multitude of contributions has created an environment in which many systems operate as standalone systems without interoperability or shared resources.

Additionally, teams need to always be concerned with all aspects of the system that are being developed (electronics, hardware programming, communication, data management, and security), or bring together a host of subsystems and convince them to work together. For instance, the building of sensor networks takes a lot of time and resources. There are only a few open-source methods for integrating various devices and sensory modalities, making it challenging to expand on prior work [26]. In response, researchers have suggested IoT designs based on the use of various architectural layers and cutting-edge technologies for the end-to-end integration of IoT systems [26]. However, the existing solutions present many challenges, such as: interoperability [27,28], scalability [23,29], and complexity in customization and extension [23].

To address these challenges, we proposed the monitoring and control framework (MCF), a conceptual structure for the design and implementation of IoT projects in multiple domains. MCF utilizes a five-layer IoT architecture reference model, focusing on scalability, security, and interoperability. We have released our code for use, and to obtain open-source contributions from the IoT community, at https://github.com/dbms-ku/iot-mcf (2 January 2023). We demonstrate the effectiveness of the framework through its implementation in a case study looking at one of the targeted domains: smart agriculture. Our results show that MCF is a valuable tool for ensuring the success of IoT projects in a variety of industries. Our contributions can be summarized as follows:We propose an end-to-end IoT solution to monitor diverse phenomena using sensors. Our solution has extensibility, scalability, and interoperability as its main advantages, and allows for users to easily create and tailor our solutions to their needs;Our solution parts from the principle that the user wants to start his project straight away, providing the tools for a rapid prototyping, construction, and testing;Our solution is focused on using commercial, off-the-shelf products, which makes it much cheaper than commercially available systems. A price comparison with common solutions is provided;We provide an open-source code for our framework so that the user can use it as necessary;We describe our real-world use-case and provide some steps for the utilization of our framework.

The remainder of this paper is organized as follows: the next section, Section 2, presents a review of the relevant literature on the existing frameworks for the Internet of Things (IoT), as well as the motivation, objective, and scope of this work. Section 3 discusses the five-layer architecture that forms the foundation of our framework and provides a conceptualization of the monitoring and control framework (MCF). Section 4 provides a detailed overview of the problems that users often find in IoT systems and how they are addressed using the MCF. Section 5 describes the MCF’s deployment in a smart agriculture monitoring system. Section 6 presents a discussion of the results and implications of our implementation. The final section, Section 7 summarizes the key points of the paper and highlights future research directions.

## 2. Background

Before introducing our framework, we establish a parallel between our work and the related works in Section 2.1. We then explain the motivations that led us to develop the framework in Section 2.2. We finish this section by stating the objectives and scope of this work in Section 2.3.

### 2.1. Related Work

IoT adoption has consistently grown over the years, with many authors noting its use in the provision of solutions across various industries [3,16,26,30,31]. As such, the development of IoT architectures and related frameworks remains an important topic of study. There are recognized challenges in the adoption of solutions, such as concerns regarding a lack of standardization [17,32], connectivity issues [28], and security and privacy [33,34]. Palit [35], Kakkar et al. [36], and Mirani et al. [26] provide overviews of the various IoT architecture frameworks, including their components and functionalities, further corroborating these challenges.

Various researchers have developed application-specific architectures for IoT [37]. Many of these architectures are domain-specific, with a few of them being used for multiple domains. Khaoula et al., propose an architecture for an aquaponics system powered by solar energy [38], Quy et al. [39] and Aivaliotis et al. [40] propose an architecture for healthcare, Kniess et al. [41] and Salazar-Cabrera et al. [42] propose an architecture for transportation, and Coito et al. [43] and Voicu et al. [44] propose an architecture for industry. Despite the great results that were achieved in these works, they are all domain-specific and not applicable to other domains, or they require extensive modifications to be applied to other domains. This makes the integration of different solutions difficult.

Konduru et al. [45] clearly states that IoT has potential regarding the building of cross-domain IoT frameworks and tools, which will allow for new and unanticipated applications and value-added services. In this regard, there have been works proposing cross-domain architectures. Trakadas et al. [46] proposed a domain-agnostic reference architecture that is capable of supporting heterogeneous devices in various network environments. Similarly, Sun et al. [47] and Neto et al. [48] propose architectures for cross-domain-sharing capabilities. However, these and many of the other proposed cross-domain and domain-agnostic architectures do not provide end-to-end implementation assistance in terms of software code. The focus of IoT frameworks, however, is addressing challenges in specific IoT domains [49]. As a result, there are major inadequacies in the literature and practice in terms of framework architectures that are both domain-agnostic and present an implementation code.

To answer this, some authors propose domain-agnostic IoT architecture frameworks with code. Piadyk et al. [50] introduce a reconfigurable environmental intelligence platform (REIP) for fast sensor prototyping, providing a software framework that can be implemented in Python. REIP provides an SDK that includes dozens of blocks for commonly required tasks such as data acquisition, processing, and storage. REIP SDK is presented to alleviate the engineering burden of implementing sensor networks. However, REIP requires the sensing device to connect directly to the cloud, which is not feasible in locations with poor or no internet coverage. REIP also does not provide for IoT systems that control actuators in addition to monitoring. The requirement that sensing devices connect to the cloud, and the lack of support for control through actuators are common features of the multi-domain architectures that provide a software framework, as observed with the Signpost Platform [51] and SensorCentral [52]. Cloete et al. [53] propose a system architecture for sensing and control without an end-to-end software framework for quick prototyping.

To the best of our knowledge, there is no domain-agnostic IoT architecture for both monitoring and control, with a software framework that is suitable for implementation in locations without internet coverage. Therefore, we propose a monitoring and control framework that is domain-agnostic and has subsystems for both monitoring and control, as well as a computing subsystem that is self-sufficient without connecting to the internet, making our framework suitable for locations without internet coverage. The MS and CTS of the MCF connect to CPS, which can optionally connect to the internet, reducing the entire system’s exposure to the public internet for enhanced security.

Our framework is a first step toward the domain-agnostic standardization of shelf sensors. We follow the two standardization strategies proposed by Motlagh et al. [29]: First, we define systems using a shared understanding to enable equitable access and usage by all stakeholders. Secondly, we create open information models, architectures, and protocols for the standards that are freely and publicly available. Our proposed end-to-end IoT monitoring and control framework and implementation code provides the building blocks used to create IoT architectures and can be used in a wide range of scenarios, regardless of the domain.

### 2.2. Motivation

The MCF was created to address IoT challenges such as scalability, lack of standardization, connectivity, reusability, and security and privacy. The MCF seeks to provide a domain-agnostic solution to these challenges to address the lack of standardization and encourage reusability.

### 2.3. Objective and Scope

The objective of this study is to design and develop the MCF to improve the scalability, interoperability, and security of IoT projects in a domain-agnostic manner while reducing development and maintenance costs. The MCF aims to provide an end-to-end solution for monitoring and control in IoT. The scope of the MCF includes the design and implementation of the monitoring subsystem, control subsystem, and computing subsystem, which work together to collect and process data and control devices. The MCF provides a flexible and extensible architecture, allowing for it to be adapted to a wide range of applications and domains. The MCF is also accompanied by a code for implementation, to speed up development and reduce the associated costs.

## 3. MCF Conceptualization

Several different reference models and architectures have been proposed for the Internet of Things (IoT) by different organizations, such as CREATE-IoT [54], OneM2M [55], IoT-A [56], and FIWARE [57]. These models differ in the number of layers and their respective functions, with the most common being the three-layer, five-layer, and seven-layer IoT architectures. The three-layer IoT architecture consists of the following layers [27,58]:**Perception/Sensing Layer (PSL)**: This layer includes the physical devices and sensors that collect and transmit data from the physical world;**Transportation/Network Layer (TNL)**: This layer includes the communication infrastructure that connects the devices and sensors to the platform and enables data transmission;**Application Layer (APL)**: This layer includes the applications and services that run on top of the IoT platform and enable users to interact with the devices and sensor data.

The five-layer IoT architecture adds two additional layers to the three-layer model:**Perception/Sensing Layer (PSL)**: This layer has the same functionality as the three-layer model;**Transportation/Network Layer (TNL)**: This layer includes the hardware and software infrastructure that supports the IoT devices and sensors, as well as the communication protocols and data management systems;**Middleware/Processing Layer (MPL)**: This layer includes the software and services that provide functionalities such as data analytics and visualization;**Application Layer (APL)**: This layer has the same functionality as the three-layer model;**Business Layer (BSL)**: This layer includes the business processes and applications that leverage the data and services provided by the IoT platform to achieve business objectives.

The seven-layer IoT architecture adds two additional layers to the five-layer model:**Physical Devices and Controllers Layer**: This layer includes the physical devices and sensors that collect and transmit data from the physical world;**Connectivity Layer**: This layer includes the hardware and software infrastructure that enables the devices and sensors to communicate with each other and with the rest of the IoT system;**Edge Computing Layer**: This layer includes the hardware and software infrastructure that supports edge computing, which refers to the processing of data at or near the source of data generation rather than in a centralized location;**Data Accumulation Layer**: This layer includes the data storage and management systems that store and process the data collected by the devices and sensors;**Data Abstraction Layer**: This layer includes the software and services that provide functionalities such as data analytics and visualization;**Application Layer (APL);****Collaboration and Processing Layer**: This layer includes the business processes and applications that leverage the data and services provided by the IoT platform to achieve business objectives.

In Figure 1, the layers are arranged from top to bottom depending on how close or far each layer is from the human level or the hardware level. Each of these reference models has its own strengths and weaknesses, and organizations may choose to adopt a particular model based on their specific needs and requirements. The five-layer IoT architecture reference model provides a middle ground that expands on the functionalities of the three-layer model, while adequately covering the granularity of the seven-layer model [27]. Thus, the five-layer IoT architecture is used as a base reference model to streamline the functionality of the MCF by carefully considering the responsibilities of each layer of the reference model. This approach ensures that our proposed framework provides a holistic implementation, which is capable of being deployed as a complete IoT system. In the following subsections, we will describe the types of solutions provided by our framework to solve common problems with each layer.

### 3.1. Perception/Sensing Layer (PSL)

The Perception/Sensing Layer, also known as the device layer, is the lowest layer in the IoT architecture. This layer is responsible for generating and collecting data using sensors, as well as affecting change in the environment or in systems with the use of actuators.

Sensors are devices that detect changes in a physical property (such as temperature, pressure, or light) and convert these changes into an electrical signal that can be measured and analyzed. Many different types of sensors are available, including vision and imaging sensors, distance/position sensors, pressure sensors, gas sensors, radiation sensors, temperature and humidity sensors, flow sensors, and contact sensors. These sensors serve different purposes, including electrical circuit monitoring, weather monitoring, chemical monitoring, and optical monitoring. Many different types of sensors are available, and they can be broadly classified as digital or analog. Digital sensors report data in discrete values, while analog sensors report data in continuous values. The type of sensor that is used will determine whether digital or analog input/output pins on the microcontroller board are used to connect the sensor. It is important to choose an appropriate type of sensor for the data that are being collected, based on the accuracy and resolution that are required, as well as the range of values being measured.

Actuators convert an electrical signal into a specific physical action. These can be classified by the source of motion energy (such as electrical, hydraulic, pneumatic, thermal, or magnetic) or the type of motion that is produced (such as rotary, oscillating, linear, or reciprocating). Actuators are used to turn command instructions into precise actions, causing a change in system status, the environment, or phenomena.

In addition to sensors and actuators, the PSL includes microcontrollers that are used for hardware communication and to perform basic transformations on the data generated by the sensors, or to decode and encode control instructions. Thus, it is typically considered to be the foundation of any IoT system, as it is responsible for providing the data on which the entire IoT system depends. The number of sensors and actuators needed in this layer depends on the system objectives and the type of data being collected. The selection of a microcontroller also plays a key role in the scalability of an IoT implementation, as they only support a limited number of input/output pins, which limits the number of modules, such as sensors, actuators, and devices, that the microcontroller can effectively control under normal operating conditions. In cases where the number of modules that are needed is beyond a ability of a single microcontroller, multiple microcontrollers can be combined into an integrated system using a digital interface such as the Recommended Standard 485 (RS-485), Inter-Integrated Circuit (I2C) or Controller Area Network (CAN) bus.

Given that sensors are particularly prone to providing intermittent erroneous values, some level of data-processing occurs at the PSL to ensure that data are correct and complete. Our framework provides initial data processing functionalities, as explained in the subsections below. Energy consumption and some special considerations regarding actuators are discussed in the following subsections.

#### 3.1.1. Error Detection and Correction

The error detection and correction step during the initial data processing step is important in ensuring the accuracy and reliability of the data that are collected and transmitted. In this step, our framework selects any data values that are outside the expected range for a particular sensor. These are identified as errors and replaced with the maximum or minimum acceptable value. This helps to eliminate any incorrect or misleading data that could impact the performance or functionality of the IoT system. Using this error detection and correction method, the data recorded for transmission are structured to suppress any observable deviations. Equation (Equation 1) presents a mathematical representation of the error detection and correction.
(1)$$V_{reg}=\left\{\begin{array}{cc} x_{max} & x>x_{max}\\ x & x_{min}\leq x\leq x_{max}\\ x_{min} & x<x_{min} \end{array}\right.$$
where $V_{reg}$ is the regularized sensor value, *x* is the reported sensor value, $x_{min}$ is the minimum reasonable sensor value, and $x_{max}$ is the maximum reasonable sensor value. For every sensor, variables for $x_{min}$ and $x_{max}$ are provided for configuration to ensure that they suit the use-case.

#### 3.1.2. Data Smoothing

The simple average is used as a data-smoothing technique to reduce the impact of noise or fluctuations in sensor data. This is especially useful when the sensor is sensitive to external factors or when the data being collected may contain sudden spikes or jumps. By taking the average of multiple sensor readings, our framework can smooth out these fluctuations and obtain a more stable and reliable representation of the data. This is particularly useful for long-term data collection and analysis, where sudden spikes in the data could lead to incorrect conclusions or misleading results. Equation (Equation 2) represents the average of the reported sensor values
(2)$$V_{av}=\frac{1}{N}\int_{0}^{N}s\left(t\right)\;dt$$
where $V_{av}$ is the average value; *N* is the number of reported values; s(t) is the sensor function over time *t*. For every sensor, we provide a variable that allows for *N* to be configured to determine how many values should be averaged. If spontaneous spikes are considered reasonable or valid data, *N* should be set to 1.

#### 3.1.3. Data Transformation

Typically, different sensors report values from different value ranges. Some might be larger than others. As such, the required precision of each sensor may be different. To avoid losses in data precision, our framework transformed the averaged value into an integer so that the inverse of the transformation was performed on the transmitted data to recover the averaged value. Equation (Equation 3) shows the mathematical representation of the data transformation function
(3)$$V_{trans}=lv+k$$
where $V_{trans}$ is the transformed value, *l* is the multiplicative function, *k* is the additive constant, and *v* is the averaged value. For every sensor, we provide variables for *l* and *k* to be configured to define the transformation function. Where such a transformation is not desired, *l* should be set to 1, and *k* should also be set to 0.

Figure 2 summarizes the initial data processing that was carried out at the PSL before the data were transmitted to the CPS. Readings from a sensor go through error detection and correction, data smoothening, data transformation, and data packaging to generate the reported value.

#### 3.1.4. Energy Consumption

Energy consumption is a critical concern in IoT systems. To minimize energy consumption, it is paramount that the systems designed in this layer adhere to a well-planned power management scheme. One such scheme is the sleep-and-wake cycle, where sensors are turned off during their inactive states and only activated to record readings. Each sensor in a device maintains its own sleep-and-wake cycle to ensure that every sensor is independent of the others, as shown in Figure 3b. With the sleep-and-wake cycle, in the sleep state, the sensor is turned off and consumes little to no energy until the end of the sleep mode. These cycles are precisely controlled by the microcontroller in a systematic loop, as shown in Figure 3a. The framework provides a robust scheme, with time-based or CPU clock cycle-based configurable variables that determine the sleep mode and wake mode of all modules connected to the microcontroller. The flowchart shown in Figure 4 describes the power management scheme as designed in the MCF. For each sensor cycle, calculations are only performed if the state of the sensor allows this. This state is based on pre-determined energy-saving profiles that allow for the user to control how much energy is being spent on the system.

#### 3.1.5. Actuators

The use of actuators presents a different challenge. They are typically expected to precisely respond to control instructions, provide periodic status reports and, on occasion, respond to user requests for system status checks and other predetermined functionalities. Thus, the data processing requirements differ slightly from the case of sensors and other modules. As expected, the MCF allowed for initial data processing before the instructions were executed, and for response protocols. The initial data processing that is provided specifically for control systems by the framework includes:**Signal Decoding**. The received signals require decoding because these signals are encoded before transmission. The structure of the message payload is similar to Figure 5. In this case, the data component of the received message requires decoding;**Signal-to-Instruction Mapping**. Here, there is an attempt to map all decoded signals to a corresponding instruction. Any signal that is not successfully mapped to an instruction is simply dropped or ignored. Three control instructions are supported by default by the MCF, with the ability tAmeno easily extend the instruction set to meet any project’s specifications. The three default instructions are: (1) checking the actuator status, (2) turning the actuator on, and (3) turning the actuator off;**Instruction Execution**. This is the point where the actuator performs the instruction, decides whether to turn on or off, or checks its status and sends the report;**Status Response**. The execution of all valid instructions is followed by a status check-and-response procedure. The current status of the actuator is recorded, packaged, and transmitted.

Figure 6 summarizes the processes that signals go through, from signal decoding, signal-to-instruction, to instruction execution, with the status response shown in the block diagram.

The processes described above (error detection and correction, data smoothening, data transformation, and data packaging), together with the sleep-and-wake cycle, determine sensor behavior in relation to data management. To enhance the reusability, extensibility, and readability of the instructions running on the microcontroller at this layer, we designed the code in a modular fashion. In view of the fact that IoT projects have different requirements and objectives, we designed the code for the framework to make it easy to add/remove sensors to/from the project. We created a modular sensor library that makes it easy to work with sensors or modules that are not yet included in the library. Figure 7 shows the class diagram of our modular sensor library, which includes 11 sensor classes derived from the parent class, the RegressiveDataTransformer class (derived from the DataTransformer class) for transforming the sensor data, and the SensorHandler class, which is responsible for managing the sensors’ sleep-and-wake cycles.

### 3.2. Transportation/Network Layer (TNL)

This layer, sometimes called the platform layer, is responsible for data transmission and connectivity between the different subsystems, end nodes, and modules of an IoT system. It also handles the routing of data to and from the cloud infrastructure that manages IoT services.

Several communication technologies and protocols are available for use in the TNL, each with different capabilities, and each suitable for different use-cases [59,60,61]. These technologies can be classified into four main categories: device-to-device (D2D), device-to-application (D2A), device-to-gateway (D2G), and device-to-cloud (D2C) [62].

During D2D communication, devices establish direct communication with each other, without the need for an application server, base stations, or access points. D2A communication involves seamless communication between IoT devices and applications through well-implemented protocols. In D2G communication, devices communicate through a local gateway, such as access points and network servers, which act as a middleware service provider for communication translation. Finally, in D2C communication, devices directly handle information transfer and resource management on a cloud service infrastructure.

Different communication technologies and protocols may be combined in the TNL to control the flow of messages and optimize throughput, power consumption, and resource usage. The TNL plays a critical role in the IoT system, as it enables the devices and sensors to communicate with each other and other system components.

#### 3.2.1. Communication

We implemented the framework with LoRa as the radio communication module, providing an easy modification that allows for the use of other communication modules, such as nRF24L01. LoRa is a long-range modulation that can cover regions up to tens of kilometers away in rural areas and a few kilometers away in urban areas [63]. In comparison to competing technologies, the LoRa SX127x family from Semtech Corporation provides significant benefits in terms of range, reliable performance, and battery longevity [64].

The MCF provides three options for communication:**Acknowledged Message**. This option is a bare-bones option that automatically supports the acknowledgment of messages sent, and executes a number of retry attempts in the event of failure. The number of retries is configurable;**Round Robin Communication**. This option places an extra layer of functionality over the acknowledged message option such that each device is assigned a time slot to offload message payloads. We provide configuration variables that can determine the minimum time interval between successive transmission opportunities;**Multi-receiver Communication**. This provides extra functionality to support communication with a large number of devices. This option involves the use of multiple receivers on the same central node to coordinate, receive, and send acknowledged messages.

A receiver, in the context of our implementation, is a module principally made of an Arduino microcontroller and a LoRa module, which is programmed to autonomously receive and send messages and uses serial output for communication with other devices. The receiver uses the Ack Code in the message to acknowledge receipt of the payload to the sender. After a successful acknowledgment process, the payload is unpacked to extract the data component. The data component is written to a serial output. This ensures that the receiver can easily be replaced by any module with a serial output, thus making the system customizable.

#### 3.2.2. Data Packaging

We also implemented communication acknowledgment. Message failure leads to retries to increase successful message delivery in challenging environments. Each message that is to be transmitted over the network has a randomly generated alphanumeric token called an Ack Code attached to the data payload, which is used to authenticate and confirm the receipt of the transmitted payload. The configurable parameters for efficient communication include the token’s size, the alphanumeric characters that are permitted, and the amount of time until a communication exchange is classified as unsuccessful. This offers the flexibility to avoid situations such as data collisions. Equation (Equation 4) was used to calculate the minimal length of the Ack Code *N* for any character set size *S* and the number of devices *D* with an Ack Code collision probability of 1 in 100,000 (that is, 0.00001), as well as a worst-case scenario in which all devices on the network are communicating simultaneously:(4)$$\frac{1}{100,000}=\frac{D}{S^{N}}$$

We rewrote Equation (Equation 4) to determine the minimum recommended Ack Code size, as shown in Equation (Equation 5):(5)$$N=\frac{5+log(D)}{log(S)}$$

In terms of payload configuration, the MCF uses a fixed payload size. This enables messages to be sent successfully without reserving a special character as a delimiter to mark the end of a message, and also ensures uniformity and standardizes the data unpacking process. We provide a configuration variable to control the payload size, allowing for the size to be customized to suit project specifications. The structure of the payload consists of three components: the acknowledgment code, the message sender’s address, and the data, as shown in Figure 5.

### 3.3. Middleware/Processing Layer (MPL)

The Middleware/Processing Layer, also known as the Data Layer, serves two essential functions in an IoT architecture. First, it acts as the data accumulation layer by aggregating data from all sources and managing the flow of data and control instructions. This requires the MPL to be able to accept and interpret different communication protocols, data formats, and types. To ensure interoperability, the design of this layer should consider syntactic interoperability (allowing for different types of applications to communicate and share data regardless of their language or protocols), structural interoperability (ensuring data exchange formats are standardized and homogeneous), and semantic interoperability (ensuring that the meaning of exchanged data and information is preserved).

The second function of the MPL is to process the received data and store them for future retrieval for reporting and analytical purposes. Security and privacy are crucial in this layer, as vulnerabilities may lead to compromises in the data, which could adversely affect the usability of upstream data. Scalability and reliability are also important considerations in this layer, as the system should be able to handle increasing data flows and maintain availability.

Two common technologies used for data storage in IoT projects are relational databases, such as MySQL and PostgreSQL, and NoSQL databases, such as MongoDB and Cassandra [65,66,67]. These technologies offer different trade-offs in terms of performance, scalability, and data model complexity, and the appropriate choice depends on the specific requirements of the IoT system. The MPL plays a critical role in the IoT system, as it enables the aggregation, processing, and storage of data from the devices and sensors.

The MCF provides data processing and decision-making functions, in addition to data modeling and storage. We implemented the MPL using Flask, a micro-web framework written in Python. As it does not require any specific tools or libraries, it is considered a micro-framework. Flask has no database abstraction layer; as such, we used MongoEngine (as a document–object mapper) and MongoDB to store processed data. MongoDB is a cross-platform document-oriented NoSQL database program that uses JSON-like documents for data storage. The data model implemented with MongoEngine is shown in Figure 8, with six documents: Sensor, SensorData (individual records of data from sensors), Device (collection of sensors on a common Arduino), Field (group of devices in a physical location), Actuator, and ActuatorStatusChange (individual records of changes in the status of actuators).

### 3.4. Application Layer (APL)

The APL is responsible for enabling users to interact with the devices and sensor data in the system through applications and services. It sits above the MPL and below the BSL in the reference model. To facilitate this interaction, the APL applies a suitable data formatting protocol for effective data pushing and pulling. This can be implemented using various protocols, such as REST, WebSocket, Message Queue Telemetry Transport (MQTT), FTP, and HTTP/HTTPS [68]. These protocols can be used to implement different communication modalities, such as client/server architectures and subscription mechanisms.

Some protocols, such as FIWARE and MQTT, use a publish–subscribe system, which is a distributed data messaging system that efficiently handles multiple data streams by categorizing data into independently accessible classes in a centralized broker. The publish–subscribe system allows for devices to publish data to a centralized broker, which then distributes the data to subscribed devices. This can be useful when handling large volumes of data or enabling real-time communication between devices. The APL serves as an interface between the lower layers of the system and the users, translating the data and functionality provided by the lower layers into user-friendly applications and services. It plays a critical role in enabling users to access and interact with the data and services provided by the IoT system.

We provided an extensible RESTful application programming interface (API) with some initial endpoints. Representational state transfer (REST) is a software architectural style that defines a set of constraints and properties for the creation of web services. API endpoints are the specific points at which the API can be accessed by a client. By creating extensible endpoints, it is possible to allow for the expansion and customization of the API as the system scales to accommodate additional functionality. The API provides JavaScript object notation (JSON) data exchange. JSON is a lightweight data-interchange format that is easy to parse and generate, making it well-suited for use in IoT systems. There are numerous front-end technologies and libraries, such as React and VueJS, that can be utilized to provide user interactions and different teams have different preferences in terms of graphical user interfaces; hence, we do not provide a code implementation for user interfaces in the MCF, although a simple interface is provided with ReactJS in the Smart Agriculture implementation. However, we provided API endpoints that are accessible to most available front-end frameworks.

### 3.5. Business Layer (BSL)

In a five-layer IoT architecture, the BSL is the highest layer and is responsible for defining the overall goals and objectives of the IoT system. It determines the value proposition of the system and defines the customer segments that it targets. It comprises analytical, visualization, and perception services that focus on analyzing the data provided by the IoT subsystems to provide users with useful information and insights for data-driven decision-making. These services can take various forms, such as dashboards, reports, alerts and notifications, depending on the specific requirements of the system.

The BSL also handles user interactions with the IoT system by receiving control commands and user preferences through intuitive interfaces. These interfaces can be web-based, mobile-based, or other types of user interfaces that allow for users to interact with the system in a convenient and user-friendly way. It is critical to the success of an IoT system, as it determines how useful the users perceive the services to be. As a result, a careful design and implementation, which focuses on user-friendliness, responsiveness, security, and user privacy, is essential. The BSL plays a crucial role in defining the overall value proposition of the IoT system and determining its target customer segments. In general, this layer determines the overall goals and objectives of the system and defines the business processes and rules that govern its operation.

For the avoidance of doubt regarding the difference between the BSL and the APL, the BSL is focused on the business goals and end-user experience, while the APL is focused on the technical aspects of data exchange and communication between the different system components. As a result, the BSL is domain-specific and is not directly provided in the MCF. However, Section 5 demonstrates the BSL, where a farmer is able to monitor soil and atmospheric conditions, and control irrigation in paddy rice fields using a curated business logic for this specific use-case.

## 4. Monitoring and Control Framework Implementation

IoT projects, independent of the domain, are usually structured in many different subsystems. This relates to scalability and the reduction of having a single point of failure (SPOF). In this section, we demonstrate the actual implementation of our framework in terms of three different subsystems. Our objective is to provide a conceptual structure for the design and implementation of IoT projects that is drawn from experience and aims to address common IoT system challenges, such as scalability, reusability, and interoperability, that can arise from the lack of standardization in the IoT space.

Figure 9 illustrates how each subsystem exchanges information with each other, and their respective responsibility. Each of these subsystems will be explained in detail in the following subsections.

### 4.1. The Monitoring Subsystem

The monitoring subsystem is responsible for collecting (or generating) data related to a phenomenon that is being monitored. This subsystem typically consists of sensors, a microcontroller board, a communication module, and a power system supply system. The sensors convert the physical properties of the environment into electronic signals that can be measured and interpreted as data. These data are then processed before being sent to interested entities. The micro-controller board is responsible for orchestrating the whole operation: communicating with the sensors to fetch data, performing the necessary transformations to these data, packaging them and transmitting them through the communication channel. In our implementation, the communication is carried out via wireless antennas, and the entity that receives these data is the computing subsystem. Figure 10a presents a simple overview of the monitoring subsystem.

#### 4.1.1. Monitoring Subsystem Concerns

From our experience, when a user needs to build an effective monitoring subsystem, there is one important point that needs to be ensured, namely data reliability; otherwise, the whole operation can become compromised. If there is no trust that the data arriving to the server is correct, the decisions taken by other subsystems can have catastrophic results. To guarantee that the data are correct, the user needs to consider the two major SPOFs:**Data reading:** The process of reading the sensors’ value for small/starting projects, in which the main source of information to the user is the sensor’s manual on how to read values. Determining how to correctly read values from different sensors may become overwhelming to users. If this process fails, the data have no value to the receiving end; in some cases, this can even lead the system to operate in the wrong way;**Data transmission:** The communication between the sensors and the receiving part is typically one-way communication, with data being transferred from the nodes to the computing subsystem. As there are many communication options, there are many errors that can arise with the chosen communication module, for example, protocol errors, packaging errors, and configuration errors. Having a solid communication protocol is essential to obtaining an effective system.

These points require the user to spend time reading documentation and making crucial decisions. However, using our framework, many of these concerns are abstracted from the user and are guaranteed to be standardized and working. The following subsection shows how a user could use the proposed framework to build a monitoring subsystem.

Another point of concern in the monitoring subsystems is their energy consumption. Unaware users may be tempted to continuously transmit the sensor data to obtain the system status in real-time. However, some sensor values do not change this often. For instance, a weather temperature monitoring sensor does not need to transmit every 1 ms, since, under normal conditions, the temperature will not change that quickly. Additionally, the careless transmission of data will flood the data transmission media, which will cause many package collisions, and the data may never reach their destination. Lastly, data reading and transmission are power-intensive tasks, and conducting these tasks too often will quickly exhaust the power system, which will result in power outages and, consequently, compromise the whole operation. How the MCF approaches these problems is explained in the next section.

#### 4.1.2. MCF Approach of the Monitoring Subsystem

Using the notation defined in Section 3, we can state that this subsystem includes elements from both the perception layer and the transportation layer. Figure 10b shows which parts can be used to make a monitoring system using our framework. Given that the user is using our supported sensors and LoRa, we remove the user overhead regarding how to reliably read the data (as the framework already provides error correction, data smoothing, and transformation). The user does not need to worry about how to send the data (as the packaging is also provided by our framework). The user only needs to select the sensors they would like to use, instantiate the data-handler to process the data from the sensors, and instantiate a communication module. Figure 11 illustrates a small monitor subsystem code using two sensors and the LoRa communicator.

Notably, the MCF uses an algorithm to control the data-reading and transmission based on the type of sensors to which it is attached. Each sensor has a value associated with the number of cycles required between readings. Doing this not only saves energy but also reduces the number of packages routed in the network, which eases the sensor burden when re-transmitting data.

### 4.2. The Control Subsystem

The control subsystem is responsible for the execution of control functions on the elements with which it is associated. This subsystem primarily comprises actuators, a microcontroller, and a communication module. Control instructions are sent from the edge device to be executed by this subsystem. Although its main function is to control, in some cases it can double as a monitoring system by providing information on the status of the controlled elements and sensors. Similarly to the monitoring subsystem, the control subsystem usually has a micro-controller board that is responsible for orchestrating the whole operation. The main difference is that, in this system, the micro-controller not only senses but is also responsible for taking action in the associated elements. This leads to a different relationship between it and the edge module, in which the communication is now bi-directional, with the system sending and receiving data. Figure 12a illustrates a simple control system.

#### 4.2.1. Control Subsystem Concerns

The control subsystem has the same SPOFs as the monitor subsystem, and some extra SPOFs related to the actuators. This subsystem needs to receive messages to control its actuators; however, many problems arise when we need to guarantee that the edge module is aware that its message arrived safely, and to inform the interested modules of the internal state of its actuators. As many messages are exchanged, the communication channel can become busy and messages can be lost, resulting in two subsystems with incoherent information regarding the real state of affairs.

These problems incur in increments in the time required for the user to ensure that the system works properly. As users need to worry about signal encoding/decoding, signal-to-instruction mapping, instruction execution, and status response. The next subsection shows how a user could use the proposed framework to build a control system with minimal effort.

#### 4.2.2. MCF Approach to the Control Subsystem

Similar to the monitor subsystem, the control subsystem uses elements from both the perception and the transportation layer. Figure 12b shows which parts could be used to create a control system using our framework. As the user is using our supported sensors, actuators, and LoRa, we can easily create a system that coordinates the sensors and actuators, reducing the user overhead when they want to create such a subsystem. Similarly to the monitoring system, the user only needs to select the sensors and actuators that it would like to use, instantiate the data-handler to process the data that come from the sensors and from the instructions to actuators, and instantiate a communication module. Figure 13 illustrates a small code of a control subsystem using one sensor, one actuator, and a LoRa communicator.

### 4.3. Edge Computing Subsystem

The edge computing subsystem is the central location for data processing and storage. It is also responsible for making decisions that determine the behavior of the control subsystem. The computing subsystem is provided by edge computing, which can optionally be supported by cloud services. This usually consists of a computing module that communicates with both monitoring and control subsystems, periodically backing up data to the cloud. This way, the edge device offers cloud services at the network edge, which enhances the response time, bandwidth usage, efficiency, and dependability of any IoT application. The edge device also provides data aggregation before transference to the cloud [69]. This setup also ensures that exposure to the public internet is reduced, as recommended by Ref. [70], since the only point of exposure to the public internet is the connection to the cloud. Figure 9 shows the MCF with the computing subsystem as a combination of an edge device and cloud services, communicating with the monitoring and control subsystems’ devices. The computing subsystem spans four of the five architecture layers, exempting the perception/sensing layer.

#### 4.3.1. Edge Computing Subsystem Concerns

Our experience shows that the main problem of the edge computing subsystem is that it needs to coordinate the transmission and reception of data through busy media while keeping the protocols in place as simple as possible. Basic functionalities such as acknowledgment of data and the means of addressing this are not usually provided, and the user has to either give up on these or implement them from scratch.

Libraries are usually hardware-dependent, and finding one that attends to the user’s needs is not straightforward. Similar to the previous subsections, these problems take time for the user and, in some cases, may even be a deal-breaker regarding whether a user can finish something on time.

#### 4.3.2. MCF Approach to the Edge Computing Subsystem

The MCF provides solutions regarding all layers used in the computing subsystem. It provides the minimum requirements for the user to build a node that is capable of sending and receiving messages to nodes in a coordinated way. Our framework provides ACK messages as well as a round-robin system that guarantees a fair system in which nodes communicate in an orderly manner, providing an equal chance for all nodes in each subsystem to transmit/receive data. MCF also provides tools for re-transmission in case of data loss.

### 4.4. Assembling MCF Subsystems

As discussed in previous sections, the MCF comprises three subsystems: the monitoring subsystem (MS), the control subsystem (CTS), and the computing subsystem (CPS). The purpose of designing the MCF to have subsystems is to allow for each subsystem to perform its tasks independently of other subsystems. This enhances the interoperability and minimizes the implementation costs, as subsystems can easily be added to or removed from the system when necessary. Figure 14 presents a visualization of the model of the three assembled MCF subsystems. The MS and CTS have two architecture layers (TNS and PSL), while the computing subsystem has four architecture layers (BSL, APL, MPL, and TNL). All the subsystems communicate with each other at the TNL.

## 5. Case Study of the Framework in Smart Agriculture

This section presents a real-world case study for a Smart Agriculture farm. The objective of this case study is to demonstrate the use of the MCF. A more detailed report is presented by Akansah et al. [11]. The system consists of three subsystems working seamlessly to monitor the ambient and soil conditions necessary for the optimal growth of paddy rice, while maintaining an adequate water level in the paddy fields throughout the growing period. The framework provides the necessary design structure, ensuring that the system is easily scalable and maintainable. The monitoring and control subsystems, including the deployment of a test, are shown in Figure 15.

The monitoring subsystem consists of weatherproof outdoor sensor nodes collecting valuable information such as ambient temperature and humidity, soil pH, soil nitrogen–phosphorus–potassium content, and soil moisture and temperature. These nodes are capable of working continuously due to the design of the reliable power system, which is explained in detail in Section 5.2. The reliability of the sensor data and the hardware communication robustness are discussed in Section 5.3 and Section 5.4.

The control subsystem is responsible for listening for a specific command instruction from the computing subsystem and then acting on the command, either by turning the water pump it controls on or off, or by sending the current state of the water pump to the computing subsystem. It sends periodic statuses regarding its current state, not only to serve as valuable information for the end-user through the user interface but also to ensure reliable and uninterrupted communication, and assure the functionality of the subsystem. This closed feedback loop relationship ensures that commands are carried out with precision in a timely fashion.

The computing subsystem, consisting of a Raspberry Pi 4 Model B development board with Broadcom BCM2711, Quad-core Cortex-A72 64-bit SoC, 4 GB of RAM and 256 GB of storage as an edge computing device, handles the data pipeline, the custom-built data visualization, and the control application. The subsystem was implemented as described in Section 4.3, ensuring efficient communication between different services, components, and applications, with a focus on extensibility and scalability.

### 5.1. Evaluation Parameters

In this section, we describe the parameters we used to evaluate the real case study:**Power Consumption**: To evaluate the amount of energy consumed by the system, we measured the voltages of the solar panels and battery pack. This metric is important because it provides insights into how much power our framework is utilizing, and consequently measures the efficiency of our system and helps to identify areas for optimization. Additionally, power can also be consumed to assess the energy usage of individual devices within the system and determine which devices are the most energy-intensive. This information can be used to target areas for energy-saving improvements and determine the overall impact of these changes on the system;**Data Reliability**: To evaluate the accuracy and dependability of the data readings obtained from sensors, devices, and other sources within our power system. This metric is important because it is necessary to have accurate and reliable data to make informed decisions and monitor the performance of the system. Data-reading reliability can be affected by factors such as device malfunctions, network issues, and interference from other devices. A low data-reading reliability can result in inaccurate readings, which can lead to incorrect decisions and affect the overall performance of the system;**Communication Robustness**: To evaluate the ability of the communication systems already in place. Communication robustness can be impacted by factors such as network outages, interference from other devices, and communication errors. Low communication robustness can result in communication failures and disruptions, which can impact the performance and reliability of the power system.

The following subsections discuss the use-case study in detail, in light of the aforementioned evaluation metrics. Section 5.2 talks about the power system, Section 5.3 talks about data reliability, and Section 5.4 talks about communication robustness.

### 5.2. Power System Evaluation

A critical component of our monitoring subsystem is our power supply system, as the reliability of the subsystem is highly dependent on its ability to work at all times, even in unfavorable weather conditions. To achieve this, our outdoor sensor nodes were powered by a pack of 18,650 lithium–ion rechargeable batteries. These batteries have a large capacity and a higher energy density, require less maintenance, and operate in wide temperature ranges (typically −20 ${}^{\circ }$C to 60 ${}^{\circ }$C). A major drawback of these batteries is their tendency to be overcharged and over-discharged; thus, to ensure the longevity of our batteries, we incorporated a battery protection system into our battery pack. We implemented an effective but inexpensive circuit using the DW01 + battery protection IC with 8205A MOSFETs to ensure that our battery pack did not overcharge and over-discharge, and provided over-current and short circuit protection. We also incorporated an efficient power distribution and charging module using the CN3791 PWM switch-mode battery charge controller with a constant voltage and current mode. The versatility of this charging solution enabled our sensor nodes to be powered by power sources such as photovoltaic cells, power banks, and DC power adapters, while recharging our battery pack. The output voltage was regulated through two MT3608 switch-mode DC-to-DC boost converters, ensuring that a constant supply of +5 V and +12 V was channeled to our different sensor categories, microcontrollers, and communication modules.

Energy efficiency is a major concern in wireless sensor networks. The sensor nodes deployed in remote field applications must employ smart power management systems to ensure continuous operation. Common approaches include designing power-efficient circuitry and reducing power usage during the node’s idle states by inducing sleep modes. The authors in Ref. [71] introduced an energy management architecture that employs an on-board, off-chip real-time-clock configuration, to control the sleep–wake phases of sensors and a microcontroller connected to IoT remote nodes. They designed and demonstrated their proposed system using the ICARUS prototype mote. The mote features ultra-low-power ARM Cortex-M4 microcontroller, some integrated sensors, and expansion sockets for wireless interfaces and other sensors. The mote exhibited roughly 22 nA during sleep mode, which is about a 98% reduction, compared to the most power-efficient boards available. However, the mote lacks high-power rails, high-voltage power, and high current sensors. A software approach was experimented with by Ref. [72] using the SWORD algorithm to implement a wake/sleep scheme for sensor devices. The algorithm compares sensor values and determines which values are sent over the communication module, thereby reducing the overall power consumption of their system by 86.45%. However, not all sensors or electronic modules have a low-power sleep mode that can be controlled by a software implementation. An example sensor is the multi-parameter soil sensor, which has a maximum working current of 12.5 mA (at 12 V), with no sleep mode [73].

For our solution, we implemented a hardware power-switching mechanism using MOSFETs controlled by microcontroller pin outputs to control the power delivery to sensors and modules during preprogrammed sleep/wake cycles. The FQP30N06L logic-level N channel MOSFET has a drain-source voltage of 60 V and a drain current of 32 A, which is capable of switching power to most power-hungry IoT sensors and modules. It also features a typical low gate charge of 15 nC, providing a fast switching time of 15 ns. This robust solution has the flexibility that allows for it to completely disable and isolate unneeded and faulty sensors, thereby preventing unnecessary power draws and possible damage to the entire system. All the components of our circuits were carefully chosen to optimize the power conversion processes to reduce losses through current leakages and heat and ensure a longer node life.

Figure 16 shows the battery voltage (in green), and the solar voltage (in blue) for recharging the battery pack for one node during our experiment. It can be observed that there was wasted power (above the red line) from the solar panel, when the battery protection system cuts off the solar voltage, to protect the battery pack. Thus, such information is vital in the optimization of the design of the power system to reduce wastage by advising a developer/user to either increase the storage capacity of the system’s battery pack or use different specifications for an optimal solar panel.

### 5.3. Data Reliability Evaluation

An extensive literature review was carried out by Refs. [74,75,76] on crucial sensors used in smart agricultural systems (SAS). These sensors play a key role in the collection of ambient, crop, and soil conditions and other relevant data to determine the state of the crops and the necessary actions to ensure optimal plant growth. Our proposed system is designed to accommodate a wide range of sensors depending on the monitoring needs of the crops and their environment. Some basic sensors, such as ambient temperature and humidity sensors, which are needed by most SAS, are incorporated by default in the sensor nodes while provisions are made for special-purpose sensors. These sensors are connected to the nodes through weatherproof aviation plugs, which ensures the continuous delivery of power and protects sensitive signal lines between the sensors and the microcontroller from fretting corrosion and oxidation. The data from the sensors are collected by a microcontroller, which is programmed to check the accuracy of the data, and the validated values are packaged and sent to the CPS through the communication module.

To test the reliability of our processed data, we deployed redundancies in the MSes; that is, two nodes per field. The objective was to use the data values collected through these redundancies to monitor and detect inconsistencies in these data. To visualize one instance, we plotted two-week datapoints showing the ambient temperature from two nodes in a single field. As can be observed in Figure 17, the graph shows very little deviation between the recorded values. These results show that our choice of sensors, although inexpensive, provided stable and reliable recordings and this phenomenon can be observed throughout our redundant sensor pairs.

### 5.4. Communication Robustness Evaluation

The main challenge in interfacing multiple hardware systems is the communication protocol. All sensors, communication modules and actuators possess a physical medium, which they use to send and receive data, and they differ greatly in their implementation. The same type of sensor, when obtained from different manufacturers, could have different communication protocols. This means that a microcontroller is required to interface with these devices and process their communication into desired outputs. This makes the choice of a microcontroller highly dependent on the choice of device, as effective communication and control are reliant on the microcontroller’s ability to accurately interpret and transmit the desired communication signals. Microcontrollers have limited resources and communication interfaces; thus, it is imperative to obtain the holistic requirements of the system before the circuitry design and hardware acquisition.

In the case of D2D radio communication, a typical LoRa radio network is set in the star-of-stars topology. This topology consists of a central node receiving multiple messages from sensor nodes, which are spread across the fields under observation. Our scenario implements a strict fixed transmission mode, which ensures that the transmitted data are encoded with the specific configuration information of the target node. The transmitted data are received promptly, without interference. The results of this implementation can be observed through the field deployment, which lasted for 132 days, during which 1,608,143 sensor values were successfully received with close to no data losses. Through the volume of data that were collected and processed, we identified key points of improvement in our system through exploratory data analysis.

## 6. Discussion

In this section, we will discuss some of the key issues and challenges in the design and deployment of IoT architectures, specifically focusing on domain restriction, scalability, interoperability, and security.

### 6.1. Domain Restriction

It is important to address the issue of domain restriction in IoT architectures. Many existing frameworks are designed specifically for a particular domain [49]. This can be limiting, as it means that these frameworks may not be suitable for use in other domains.

To address this issue, we implemented the MCF as a domain-agnostic framework. This means that the MCF can be used in a variety of domains, as it comprises subsystems (MS, CTS, and CPS) that are common to IoT applications independently of their domains. The organization of IoT systems into MS, CTS and CPS is applicable to healthcare IoT (HIoT) [77,78,79,80], industrial IoT (IIoT) [81,82,83,84,85], Smart Agriculture [86,87], Smart Energy [58,88], Transportation [89,90,91,92], environment, waste management, and security, among others. For example, the monitoring subsystem, which includes sensors and microcontroller boards, is a fundamental component of many different IoT systems, regardless of their specific domain. By designing the MCF in this way, we made it more flexible and adaptable to a wide range of different use-cases.

Furthermore, a domain-agnostic framework can help to facilitate interoperability between different IoT systems. By using a common set of subsystems, it becomes easier for different systems to communicate and exchange data, regardless of their specific domain of application. This can help to improve the overall efficiency of the IoT ecosystem, as it becomes easier to integrate different systems and extract insights from the collected data.

### 6.2. Scalability

Scalability is an important consideration for IoT systems, as it determines the system’s ability to handle an increasing amount of data and devices without a decrease in performance. The MCF framework was designed with scalability in mind, with a focus on modularity and flexibility.

One way in which the MCF is scalable is through the use of the monitoring subsystem, which can easily be expanded by adding more devices to accommodate more sensors as needed. Each device operates independently, allowing for the system to scale up without affecting the performance of other devices. The modular design of the MCF allows for the easy addition or removal of devices in the monitoring subsystem, without affecting the overall functioning of the system. This enables the system to scale up or down according to the user’s needs.

The computing subsystem is also designed to be scalable using cloud-based servers and services. By using cloud servers, it is possible to add more server resources, such as CPU, RAM, and storage, as the demand for them increases. This ensures that the computing subsystem can handle an increase in the volume of data being transmitted from the monitoring subsystem and the number of control commands being sent from the control subsystem.

Another aspect of MCF’s scalability is its ability to adapt to changes in the business requirements or goals. The business layer can easily be modified or extended to incorporate new features or functions as needed, without impacting the lower layers of the architecture. This allows for the system to easily adapt to changing business needs and remain relevant over time.

### 6.3. Interoperability

One major challenge facing IoT systems is interoperability, which refers to the ability of different devices and systems to seamlessly communicate and exchange data. This is important because IoT systems often involve the integration of a wide range of devices and systems from different vendors, with different protocols and standards. Without interoperability, it would be difficult for these devices and systems to work together and achieve the desired outcomes.

The MCF addresses the issue of interoperability in several ways. First, it uses standard protocols and interfaces, such as HTTP and REST, for communication between the monitoring and computing subsystems. This ensures that devices and systems using these protocols can easily integrate with the MCF. Additionally, the MCF includes a flexible data formatting protocol at the application layer that allows for the exchange of data in various formats, such as JSON. This allows for the integration of devices and systems that use different data formats.

The MCF also addresses this challenge by providing a standardized and modular structure for the development of IoT systems. By dividing the system into three main subsystems (monitoring, control, and computing) and implementing a clear communication protocol between these subsystems, the MCF allows for easy integration and interoperability with other devices and systems.

### 6.4. Security

Security is a critical concern in IoT architectures, as the connected nature of these systems makes them vulnerable to attacks and data breaches. The MCF is designed to minimize the exposure of devices in the monitoring and control of subsystems in the public internet. This is achieved by implementing a centralized broker in the computing subsystem that acts as a mediator between the monitoring and control subsystems and the rest of the internet. Only subsystem devices that have been registered with the broker are recognized and allowed to communicate with the computing subsystem. This registration process ensures that only authorized devices can access the system, providing an additional layer of security.

Additionally, the communication subsystem in the MCF is designed to support secure communication protocols such as SSL/HTTPS, which encrypt the data being transmitted between devices to prevent unauthorized access or tampering. This is especially important for IoT applications that handle sensitive data such as personal information or financial transactions.

### 6.5. Cost Analysis for the Monitoring System

In this analysis, we describe the costs of our proposed MS, presented side-by-side with the costs of commercial solutions with similar components and functionality. The high cost of commercially available alternatives is one of the primary reasons why people want to build custom IoT devices [93,94]. This is important because the cost of the MS, which typically consists of many devices, could have a substantial impact on the overall system cost. Furthermore, commercial solutions often lack the control functionality, as implemented in the CTS of the MCF. These commercial solutions often rely on the direct connection of sensing devices to the cloud and may lack an edge device that is comparable to the CPS.

The MS in the agricultural use-case described in Section 5 primarily contains an Arduino nano with ATmega328P microcontroller, HC-SR04 ultrasonic distance sensor, SHT20 I2C waterproof temperature and humidity sensor, RS485 Modbus waterproof multi-parameter soil-integrated sensor, LM393 rain detection sensor, DC power system with a 3.7 V 13.6 AH protected battery pack, and 2 A DC-DC Boost Step-Up Conversion Module from 3.7 V to 5 V. The MS for the use-case with all these components cost USD 390 at the time of deployment. Commercially available alternatives from well-known providers ranged in price from USD 1000 to USD 12,000. Table 1 shows the prices of similar products from recognized vendors, and the corresponding cost of an MCF implementation with similar features. According to Table 1, the low end of commercial solutions cost 400% of the MCF implementation while the more high-end solutions cost upwards of 2200%. It is worth noting that, despite ensuring that the comparisons are similar in terms of their features, there are certain nuances that were not considered. For instance, the components used by these vendors may be of a higher quality than those used for our implementation. However, the MCF provides the flexibility to use any component. Thus, there is the possibility of developing solutions with high-end components, thereby removing the limitations of commercial solutions that have specific component catalogs as add-ons to their solutions. The software component of commercial systems often uses proprietary implementation, accompanied by user support from the vendors, which contributes to the higher overall cost of their solutions.

## 7. Conclusions and Future Work

In conclusion, the Monitoring and Control Framework (MCF) is a scalable and interoperable architecture for IoT projects that aims to minimize the challenges that arise from the lack of IoT standardization. The MCF is composed of three main subsystems: the monitoring subsystem, the control subsystem, and the computing subsystem, with communications between them.

The monitoring subsystem is responsible for the collection or generation of data related to the phenomena of interest, and comprises sensors, a microcontroller board, a communication module, and a power supply system. The control subsystem is responsible for the control of physical devices based on data received from the computing subsystem, and comprises microcontroller boards, actuators and a communication module. The computing subsystem is responsible for data storage, analysis, and visualization, and comprises a server, a database, and applications.

Our framework was shown to be stable in real-world applications, with the code not incurring a significant increase in power utilization, and could be operated using common rechargeable batteries and a solar panel. It used so little energy that the usual amount of available energy surpassed up to two times the amount necessary for the system to operate.

Our framework was also proved to be reliable by accurately reading data from multiple sensors that operate concurrently. We observed that these sensors’ data are consistent and stable, with very similar readings. This contributes to the reliability of the data that are collected and processed by the system. To further improve data reliability, the sensors are designed to send readings at a consistent rate. This can be customized to represent as many details as possible and ensure that the collected information is as accurate as possible. In addition to the use of multiple sensors, the components of our framework are designed to effectively exchange data. The data exchange process is intended to be stable, with very few lost packages (over 1.5 million packages received during the experiment). This helps to minimize interruptions in the flow of information and ensures that the data used are current and accurate. Our framework is intended to provide reliable data by utilizing multiple sensors operating in parallel, as well as a stable data exchange process. This helps to ensure that the data being collected and processed are correct, consistent, and up-to-date. Our real-world use-case demonstrated the system’s dependability, power-consumption viability, communication stability, and the overall suitability of the MCF.

We demonstrated the implementation of the MCF using an open-source code and discussed the subsystems in detail, including the five-layer IoT reference model. We also discussed the communication between the subsystems and the energy optimization techniques applied in the monitoring subsystem.

In this paper, we presented an end-to-end IoT solution, MCF, which is applicable to multiple domains. Our solution has extensibility, scalability, and interoperability as its main advantages, and allows for users to easily create and customize our framework to their project specifications. Additionally, our agricultural use-case is up to 20 times cheaper than commercial solutions. We provided an open-source code for our framework, enabling a broader collaborative development.

There are several areas for future work in the development of the MCF. One potential direction is to explore the use of different communication protocols and technologies in the TNL to improve the reliability and efficiency of data transmission. Another area of exploration is the integration of machine learning and artificial intelligence techniques to enhance the data analysis and decision-making abilities of the system. Additionally, further research will be conducted on the design and implementation of our user-friendly interfaces in the BSL to improve the usability and user experience of the IoT system. Finally, continuing efforts to improve the security and privacy of the system will be essential to ensure the integrity and trustworthiness of the system.

We are aware that the mathematical formulations of this work are mostly empirical, and a more theoretical analysis is required. However, at this time, our main objective was to have it open to the community, so that it could be tested together with other researchers. We expect the community to adopt our framework, and expand upon it, making it a strong competitor to available solutions.

Overall, the aim of the MCF, which is to provide a conceptual structure for the design and implementation of IoT projects by providing a domain agnostic framework, was achieved through the development of modular and extensible subsystems. Our framework will make it easier for IoT developers to create scalable, reusable, and interoperable solutions.

## Figures and Tables

**Figure 1 sensors-23-02714-f001:**
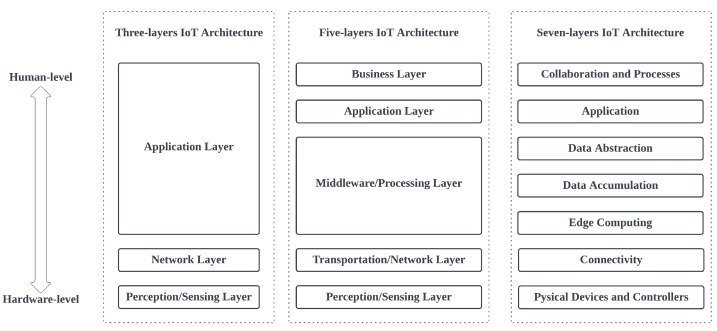
IoT architectures commonly discussed in the literature. The three-layer IoT architecture, five-layer IoT architecture, and seven-layer IoT architecture are the most commonly discussed IoT architectures in the literature.

**Figure 2 sensors-23-02714-f002:**
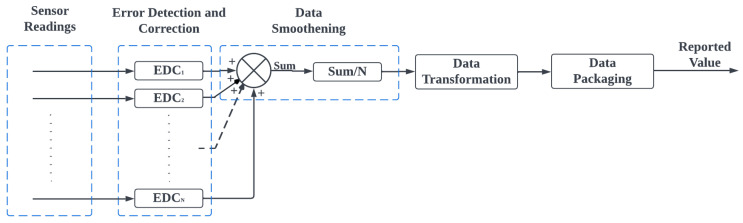
Block diagram of data processing in the perception/sensing layer (PSL). Sensor readings go through error detection and correction to regularize their values, then data smoothening, which involves taking a simple average of the regularized values, followed by data transformation and data packaging, to generate the reported value.

**Figure 3 sensors-23-02714-f003:**
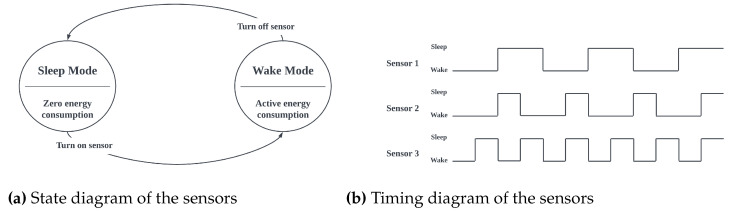
Sensors attached to a device are managed by sleep-and-wake cycles; (**a**) Sensor state diagram showing the endless sleep-and-wake cycle observed by each sensor independent of other sensors on the same device; (**b**) Sensor timing diagram showing three sensors on the same device maintaining sleep-and-wake cycles independently.

**Figure 4 sensors-23-02714-f004:**
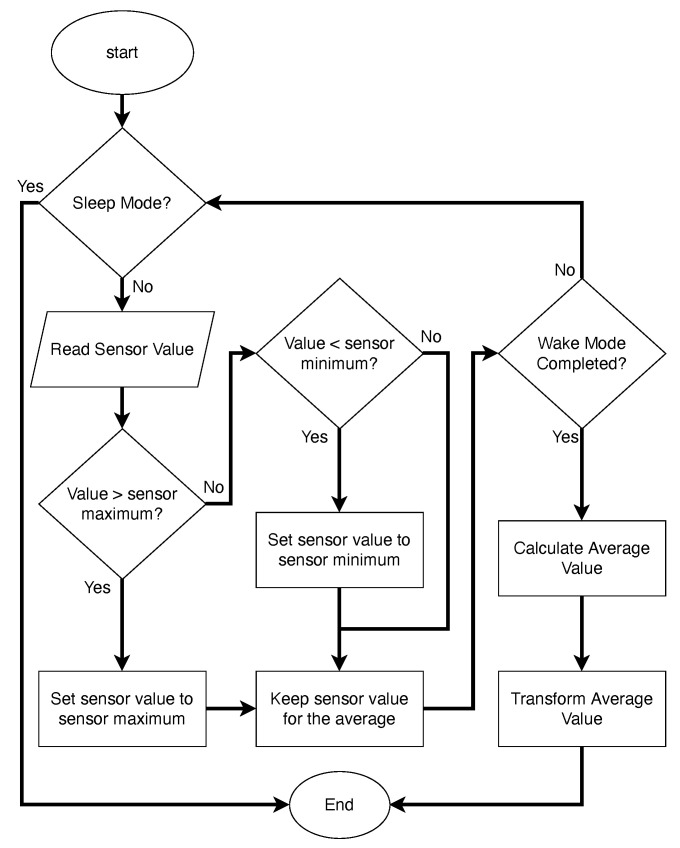
Sensor flowchart. Each sensor runs through SleepMode and WakeMode. The sensor only executes a sleep-wait in the SleepMode. The sensor loops to obtain sensor values and calculates the average when the WakeMode is completed.

**Figure 5 sensors-23-02714-f005:**
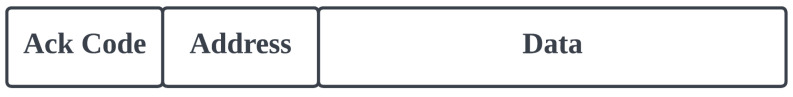
Message payload components. The message payload consists of three components: the acknowledgment (Ack) code, sender address, and data.

**Figure 6 sensors-23-02714-f006:**
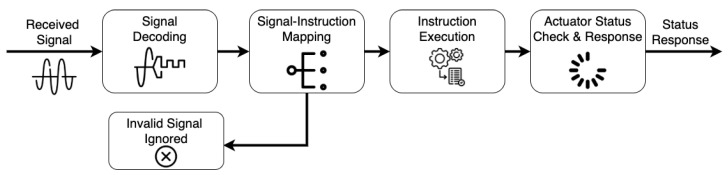
Block diagram of the processes in the perception/sensing layer. The received signals go through signal decoding, signal-to-instruction mapping, instruction execution with invalid signals ignored, and actuator status check and response.

**Figure 7 sensors-23-02714-f007:**
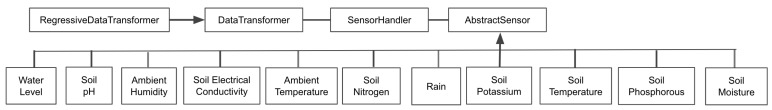
Class diagram of a modular sensor library, showing 11 concrete sensor classes derived from the AbstractSensor class, the RegressiveDataTransformer class derived from an abstract DataTransformer class, and the SensorHandler class, which is responsible for the management of the sensors’ sleep-and-wake cycles.

**Figure 8 sensors-23-02714-f008:**
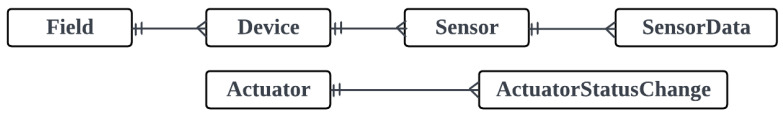
Data model implemented in the MongoDB database in the computing subsystem.

**Figure 9 sensors-23-02714-f009:**
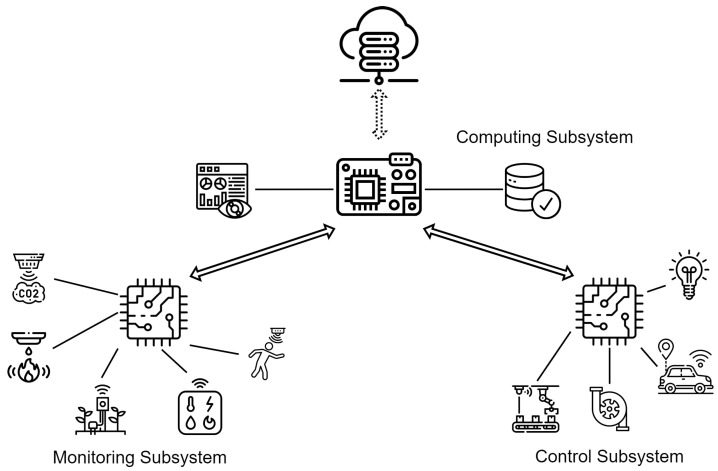
The monitoring and control framework (MCF) showing the three subsystems; monitoring and control subsystems communicating with the computing subsystem.

**Figure 10 sensors-23-02714-f010:**
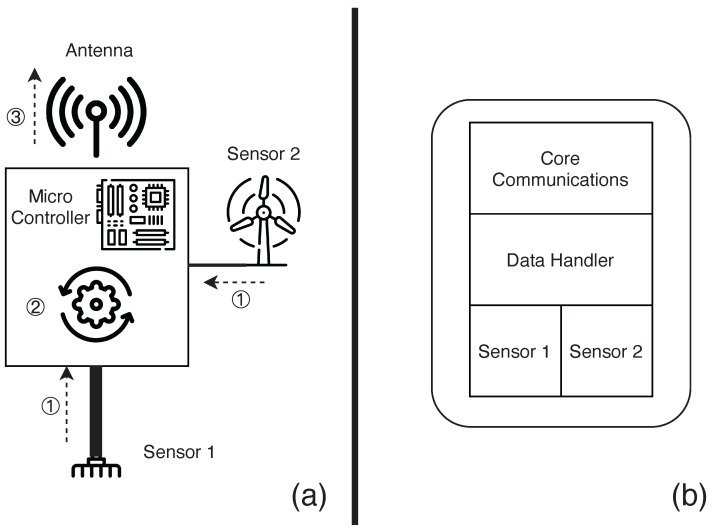
Implementation of the monitoring subsystem: (**a**) A conceptual model of a monitoring module. These data are first read from sensors monitoring a phenomenon; after that, they are internally processed in the microcontroller and finally transmitted to the point of interest. (**b**) How this module can be implemented using our framework: by using the predefined modules, fewer than 20 lines of code are needed to obtain a working system with many advantages, such as error detection, data smoothing, transformation, and packaging.

**Figure 11 sensors-23-02714-f011:**
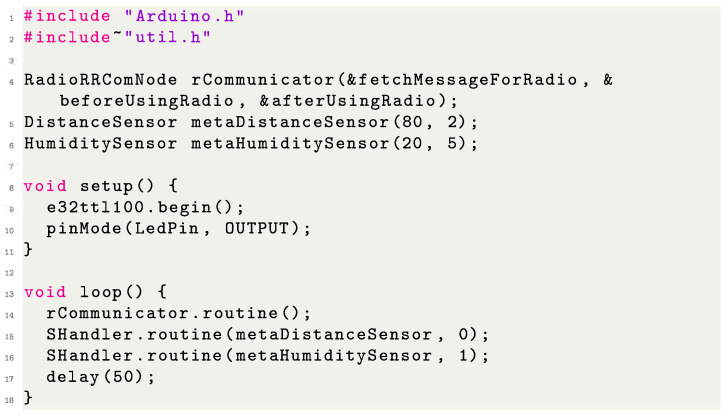
Monitor subsystem using our framework. The user needs fewer than 20 lines of code to obtain a working subsystem with all the advantages provided by the framework.

**Figure 12 sensors-23-02714-f012:**
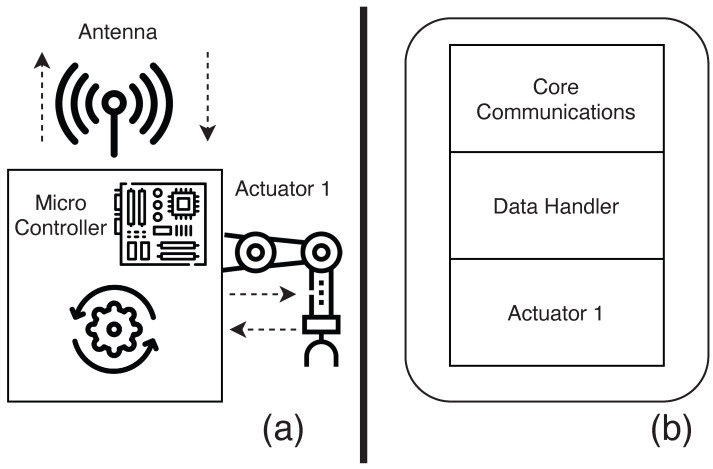
The implementation of the control subsystem: (**a**) A conceptual model of a control module. Similarly to the monitor subsystem, the data are read, processed, and transmitted to the point of interest. However, the main difference is that the module also receives instructions, which it actuates. (**b**) How this module can be implemented using our framework. By using the pre-defined modules, very little code is needed to obtain a working system with many advantages, such as error detection, data smoothing, transformation, and packaging.

**Figure 13 sensors-23-02714-f013:**
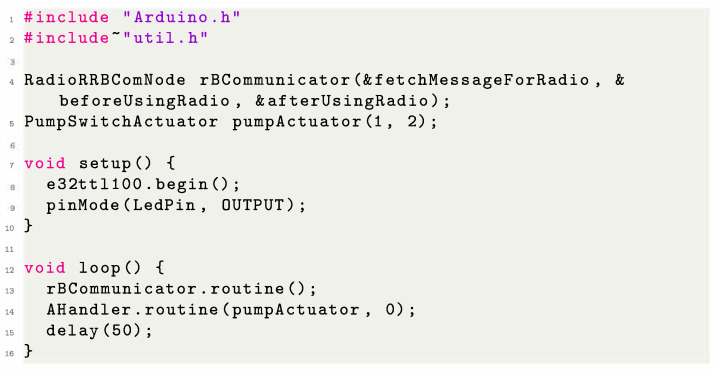
Control subsystem using our framework. The user needs fewer than 20 lines of code to obtain a working subsystem with all the advantages provided by the framework.

**Figure 14 sensors-23-02714-f014:**
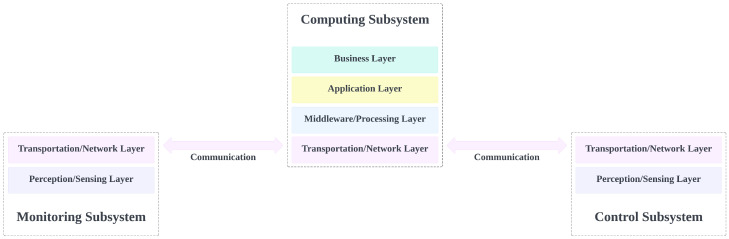
MCF subsystems assembled. The monitoring subsystem and control subsystems both have two architecture layers (transportation/network layer and perception/sensing layer), and the computing subsystem has four architecture layers (business layer, application layer, middleware/processing layer, and transportation/network layer). All three subsystems communicate with each other at the transportation/network layer.

**Figure 15 sensors-23-02714-f015:**
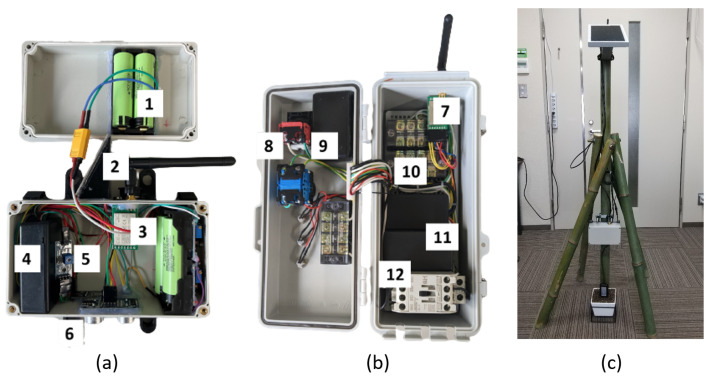
The monitoring and control subsystems of our smart agriculture use-case. (**a**) The monitoring subsystem consists of: (1) 18650 lithium–ion battery pack, (2) LM393 Rain Sensor, (3) E32-900T20D LoRa radio module, (4) Arduino nano hlATmega328P microcontroller in a protective enclosure, (5) SHT20 I2C temperature and humidity sensor, (6) HC-SR04 ultrasonic sensor. (**b**) The control subsystem, consisting of: (7) E32-900T20D LoRa radio module, (8) emergency push button and two-pole switch, (9) power supply for 5V components, (10) connection terminals, (11) Arduino nano ATmega328P microcontroller, 5 V relay and ACS712 current sensors in a protective enclosure, (12) Mitsubishi Electric S-T10 three-pole contactor. (**c**) Test deployment of the monitoring subsystem.

**Figure 16 sensors-23-02714-f016:**
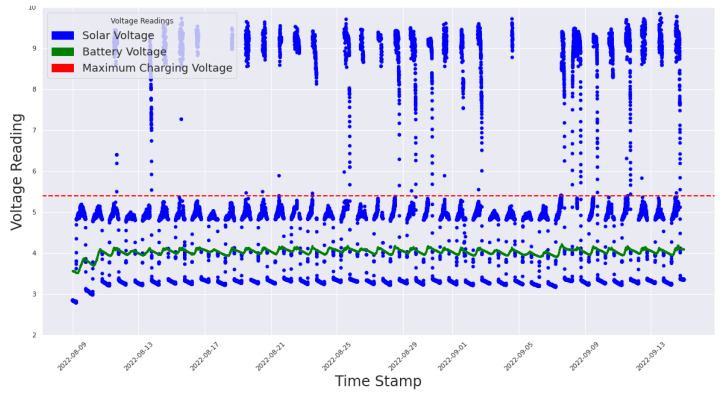
Sample visual of the recorded battery and solar voltages for one outdoor node.

**Figure 17 sensors-23-02714-f017:**
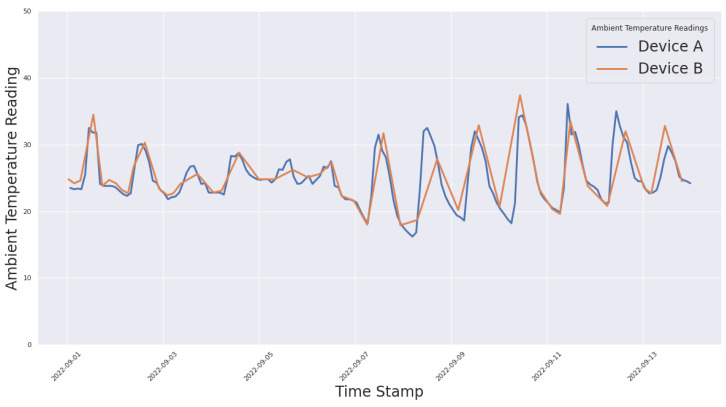
Sample visual of the recorded ambient temperature from two outdoor nodes in the same field.

**Table 1 sensors-23-02714-t001:** A price comparison between the MCF’s MS use-case and similar commercial solutions.

Vendor	Description of Product	Product Price (USD)	MCF Price (USD)	Proportion in Percentage
Vendor 1	Soil temperature and moisture	1031.84	257.85	400.17
Vendor 2	Complete Weather Station	1777.61	330.63	537.64
Vendor 3	Complete Weather Station	7447.01	330.63	2252.37
Vendor 4	Complete Weather Station	3323.92	330.63	1005.33

## Data Availability

Not applicable.

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
