# Peer review of "Monitoring and Control Framework for IoT, Implemented for Smart Agriculture"

_sensors, 2023, doi:10.3390/s23052714_

Round 1

Reviewer 1 Report

The subject of the study is a Monitoring and Control Framework (MCF) for Internet of Things (IoT) projects. The purpose of the MCF is to address challenges that arise from the lack of standardization in the IoT, including scalability, reusability, and interoperability. The MCF consists of three main subsystems: the monitoring subsystem, the control subsystem, and the computing subsystem, and is designed to be domain agnostic, scalable, interoperable, and secure. The authors also provide open-source code for the implementation of the MCF. Future work will involve testing and evaluating the MCF in real-world IoT applications.

1- The writing of the introductory part was made sloppy and left raw. Paragraph made of 2 sentences. some places have abbreviations, some don't.

2- The problem has been tried to be explained based on the literature, but can the proposed framework produce solutions to these problems? It should be presented with findings or evidence under separate headings. Although not all problems are addressed, scientific contributions should be presented clearly in substance.

3- The related works section should be reorganized to address the problems addressed by the authors.

4- Should 3 parts really be told at such length? something known to most anyone working on IoT.

5- Why isn't there a stop or terminate part in Figure 5? Even if the algorithm is an infinite loop, it must have an end. The notation/shapes used in the flow diagram drawing are used incorrectly. Please, correct it. If you can't draw, try writing pseudocode.

6- We see a box as The Control Subsystem. Do you think the reader would believe that this work was realized without seeing the internal structure? Is it an empty box?

7- The discussion and findings sections are quite weak and I don't think there is the slightest evidence to prove that the claims have been resolved. It is necessary to present this part by supporting the literature according to its contribution.

Although the current study has little scientific hope, it has major shortcomings in terms of writing, care, innovation, presentation and flow. For these reasons, I suggest it be rejected.

Author Response

We would like to thank you for the valuable comments. We have studied them carefully and have made corrections as follows.

  1. The writing of the introductory part was made sloppy and left raw. Paragraph made of 2 sentences. some places have abbreviations, some don't.

[Reply]

Thank you for your the comment. The introduction was completely revised and the problems solved.

  1. The problem has been tried to be explained based on the literature, but can the proposed framework produce solutions to these problems? It should be presented with findings or evidence under separate headings. Although not all problems are addressed, scientific contributions should be presented clearly in substance.

[Reply]

Thank you for the comment. Related works re-done and problems and solutions more explicitly written. Scientific contributions presented in substance in the abstract, introduction and related works.

  1. The related works section should be reorganized to address the problems addressed by the authors.

[Reply]

Thank you for your constructive comment. As we wrote the above, related works revised completely for clarity and linearity.

  1. Should 3 parts really be told at such length? something known to most anyone working on IoT.

[Reply]

Thank you for the comment, and we agree with the comment. Section 3 and 4 re-worked for clarity and more explicit state our contributions.

  1. Why isn't there a stop or terminate part in Figure 5? Even if the algorithm is an infinite loop, it must have an end. The notation/shapes used in the flow diagram drawing are used incorrectly. Please, correct it. If you can't draw, try writing pseudocode.

[Reply]

Thank you for your insightful comment. We agree and Figure 5 fixed with added termination and re-worked so that it shows only one cycle of a sensor.

  1. We see a box as The Control Subsystem. Do you think the reader would believe that this work was realized without seeing the internal structure? Is it an empty box?

[Reply]

Thank you for your constructive comment. We agree and the figure fixed to include more details of the system in-place as well as labels of the components.

  1. The discussion and findings sections are quite weak and I don't think there is the slightest evidence to prove that the claims have been resolved. It is necessary to present this part by supporting the literature according to its contribution.

[Reply]

Thank you for your insightful and constructive comment.  We agree and text re-worked to state more clearly our contributions.

Reviewer 2 Report

The paper title: Monitoring and Control Framework for IoT, Implemented for Smart Agriculture, is well written and nicely presented. 

The specific comments to the authors are as follows:

1.     In Abstract Section: Provide some results which shows the novelty of the work.

2. Figure 10: use proper leveling and try to show the experimental investigation. 

3. In conclusion section: Provide a point-by-point list of your major/minor recommendations for the proposed MCF for IoT. 

Author Response

We would like to thank you for the comments. We have studied them carefully and have made corrections as follows.

  1. In Abstract Section: Provide some results which shows the novelty of the work.

[Reply]

Thank you for your insightful comment. Abstract was re-worked and now contributions are more clear.

  1. Figure 10: use proper leveling and try to show the experimental investigation.

[Reply]

Thank you for your detailed comment. The figure now augmented to include more information.

  1. In conclusion section: Provide a point-by-point list of your major/minor recommendations for the proposed MCF for IoT.

[Reply]

Thank you for your constructive comment. List with major/minor recommendations included in the introduction, with statements re-enforced in the conclusion section.

Reviewer 3 Report

Author entitled “Monitoring and Control Framework for IoT, Implemented for Smart Agriculture” I would like to point out some corrections for improvements

a.      The author is advised to summarize the final outcomes in abstract and conclusion section with numerical results.

b.      In introduction and survey section should summarize the novelty or scope of this MCF and IoT for smart agriculture . Author need to add the motivation of the research and objective of this research in end of related works

c.      In fig.3 Block Diagram of data processing in the Perception/Network Layer process is look like general one. Its not related to proposed model of MCF related. Author need to modify this architecture according to the MCG.

d.      Author need to include some implementation process and environmental setup information’s. How to evaluate the performance of this systems ?

e.      Author discussed about MPL and APL in the manuscript. Author need to describe how MPL related to MCF ? and monitoring systems

f.       In Fig. 10 author pointed out Monitoring Subsystem and control Subsystem for smart agriculture. Here what is the configuration or systems set up information and cost ? what is the operating procedure ?

g.      What is the results parameters applied here ? Hoe final outcomes were evaluated ? How to visualize the final model?

h.      I never find the numerical results of the model in the manuscript. Author need to include the quantity or numerical results of the proposed model

i.       Performance validation parameter and its mathematical derivation could be included. Since the final results could be evaluated some parameters.

j.       Language correction must be done as well as grammatical errors can be avoided.

Thank you

Author Response

We would like to thank you for the comments. We have studied them carefully and have made corrections as follows.

  • The author is advised to summarize the final outcomes in abstract and conclusion section with numerical results.

[Reply]

Thank you for your constructive comment. Contributions clearer and numerical results of a cost analysis done.

  • In introduction and survey section should summarize the novelty or scope of this MCF and IoT for smart agriculture. Author needs to add the motivation of the research and objective of this research in end of related works

[Reply]

Thank you for your constructive comment. We agree and motivation and novelty are in the new re-worked introduction.

  • In fig.3 Block Diagram of data processing in the Perception/Network Layer process is look like general one. It’s not related to proposed model of MCF related. Author needs to modify this architecture according to the MCG.

[Reply]

Thank you for your constructive comment. We agree and Figure 3 is now put in context with near the relevant text to show its importance.

  • Author need to include some implementation process and environmental setup information’s. How to evaluate the performance of this systems ?

[Reply]

Thank you for your constructive comment. We agree. Real implementation is more explicit and the performance in terms of cost is provided. Also, different small problems/solution cases are now presented at Section4.

  • Author discussed about MPL and APL in the manuscript. Author need to describe how MPL related to MCF ? and monitoring systems

[Reply]

Thank you for your insightful comment. Descriptions are clearer, and are now separated into sections.

  • In Fig. 10 author pointed out Monitoring Subsystem and control Subsystem for smart agriculture. Here what is the configuration or systems set up information and cost ? what is the operating procedure ?

[Reply]

Thank you for your insightful and constructive comment. The figure improved with more information and detailed in text

  • What is the results parameters applied here ? Hoe final outcomes were evaluated ? How to visualize the final model?

[Reply]

Thank you for your insightful comment. Results in terms of costs discussed, plus section 4 makes use-cases in which the framework becomes more useful.

  • I never find the numerical results of the model in the manuscript. Author need to include the quantity or numerical results of the proposed model

[Reply]

Thank you for your insightful comment. Numerical results included in the cost analysis.

  • Performance validation parameter and its mathematical derivation could be included. Since the final results could be evaluated some parameters.

[Reply]

Thank you for your insightful comment. We added cost analysis, plus some more holistic analysis on how our framework can improve quality-of-life for users.

  • Language correction must be done as well as grammatical errors can be avoided.

[Reply]

Thank you for your comment. Text reviewed by native English speakers in the lab. If you still think that we need more improvement, please do not hesitate to point out and we would like to take editing service.

Round 2

Reviewer 1 Report

What the system does is explained in detail, but the problems or deficiencies in the literature are still not emphasized enough. The contributions presented remain at the mundane/simple level. The rest of the revisions by the authors are pretty good. In addition, the difference of the proposed system from the current/alternative/current studies in terms of discussion should be presented parametrically (especialy linked to contribution - for example, communication protocol, network structure, sensor types, power supply, data compression ratio, etc.) as a comparison table.

Author Response

Thank you for your comments, we hope we addressed them with the following changes:

  1. The literature is more emphasized and the contributions are more explicit.
  2. We added a new section commenting our data in the implementation.
  3. We also compared the costs of a system implemented using our system, versus other systems. Since there are numerous frameworks that only do specific parts, it is hard to make a fair comparison functionality-wise. We tried to be very generic and open-source, and at the same time tried to support cost-effective commercially available sensors. Most of the frameworks we found either uses very expensive hardware in a closed environment (hence we made a cost comparison for systems with the same goal), or they are too abstract that they provide a higher-level tool for you to make your solution, but you still would need lots of coding. Our framework parts from the principle that a user is willing to start immediately and wants to do it as straightforward as possible.
  4. We made a cost comparison which allows us to compare the cost of having a system built using our framework against commercially available ones.

Reviewer 3 Report

Author did not addressed my comments properly. I am not satisfied with the author response and revision. 

Author Response

Thank you for your comment and sorry for not meeting your expectations. We hope we addressed your concerns properly this time. To that, we re-reply to all of the previous comments.

Comment a: The author is advised to summarize the final outcomes in abstract and conclusion section with numerical results.
[Reply]
Thank you, indeed the abstract quality was lacking. Hence, the Abstract was completely rewritten, emphasizing more on our contributions and explaining the numerical evaluations that occurred. 

Comment b: In introduction and survey section should summarize the novelty or scope of this MCF and IoT for smart agriculture . Author need to add the motivation of the research and objective of this research in end of related works
[Reply]
Thank you, we improved the introduction even more to make it even more explicit our motivations and objectives.

Comment c: In fig.3 Block Diagram of data processing in the Perception/Network Layer process is look like general one. It’s not related to proposed model of MCF related. Authors need to modify this architecture according to the MCG.
[Reply]
Thank you, this part was modified exactly as proposed, so the authors’ believe that the dissatisfaction of the reviewer was not related to this figure.

Comment d: Author need to include some implementation process and environmental setup information’s. How to evaluate the performance of this systems?
[Reply]
Thank you, we expanded our real-world implementation section to contain more information.

Comment e: Author discussed about MPL and APL in the manuscript. Authors need to describe how MPL related to MCF? and monitoring systems.
[Reply]
Thank you, the application of the framework is more discussed now.

Comment f: In Fig. 10 author pointed out Monitoring Subsystem and control Subsystem for smart agriculture. Here what is the configuration or systems set up information and cost? what is the operating procedure?
[Reply]
Thank you again, a more robust cost analysis was included. We asked vendors and re-configure our nodes to be equal functionality-wise for a fair comparison. We then qualitatively and quantitatively compared different solutions against ours.

Comment g: What is the results parameters applied here? Hoe final outcomes were evaluated? How to visualize the final model?
[Reply]
Thank you, a longer discussion of our implementation is discussed in more details, with a new diagram included for clarity.

Comment h: I never find the numerical results of the model in the manuscript. Author need to include the quantity or numerical results of the proposed model
[Reply]
Thank you, the cost analysis was expanded and we included numerical results of our observations, with points of improvements discussed.

Comment i: Performance validation parameter and its mathematical derivation could be included. Since the final results could be evaluated some parameters.
[Reply]
Thank you, we added an analysis in which our validation parameter is the “system’s functionality” compared to other commercially available systems, taking in consideration “cost” of these systems. We contacted vendors and compared the price of them, having in mind that their solution is more robust. These tradeoffs are included in the revised version of the text.

Comment j: Language correction must be done as well as grammatical errors can be avoided.
[Reply]
Thank you for your comment, Automatic grammatical and spelling checkers were used to ensure the quality of the text.

Round 3

Reviewer 1 Report

Paper can be accepted in present form 

Author Response

Thank you for your patience and insightfull comments. We learned a lot from it and we deeply appreciate your time.

Reviewer 3 Report

Author improved their manuscript in better way however still author need to address the following comments properly without fail
The author is advised to summarize the final outcomes in abstract and conclusion section with numerical results.

In introduction and survey section should summarize the novelty or scope of this MCF and IoT for smart agriculture . Author need to add the motivation of the research and objective of this research in end of related works

 In fig.3 Block Diagram of data processing in the Perception/Network Layer process is look like general one. It’s not related to proposed model of MCF related. Authors need to modify this architecture according to the MCG.

 What is the results parameters applied here? Hoe final outcomes were evaluated? How to visualize the final model?

Performance validation parameter and its mathematical derivation could be included. Since the final results could be evaluated some parameters.

Language correction must be done as well as grammatical errors can be avoided.

Author Response

Reviewer Overall Comment: Author improved their manuscript in better way however still author need to address the following comments properly without fail:

Response: Thank you, we hope that our recent modifications can meet your expectations.

1)The author is advised to summarize the final outcomes in abstract and conclusion section with numerical results.

Response: Thank you for your comment, we added the final outcomes in the abstract and in the conclusion section. We added the numerical results we have as well.

2) In introduction and survey section should summarize the novelty or scope of this MCF and IoT for smart agriculture . Author need to add the motivation of the research and objective of this research in end of related works

Response: Thank you for your comment. Novelty is was already summarized by the end of the introduction, so we guess the remaining comment relates to the motivation and objective of the work. Now, at the end of section 2, we have two new sub-sections: "2.2 Motivation" and "2.3 Objective and scope". We hope this sections increases the readability of the paper. 

3) In fig.3 Block Diagram of data processing in the Perception/Network Layer process is look like general one. It’s not related to proposed model of MCF related. Authors need to modify this architecture according to the MCG.

Response: Thank you for pointing it out. The figure 3 is completely re-done with more details and icons to make it more intuitive.

4) What is the results parameters applied here? Hoe final outcomes were evaluated? How to visualize the final model?

Response: Thank you for your comment. We have a new sub-section (5.1) that talks about the evaluation parameters. We then proceed to discuss how each part of the real implementation is discussed in the following subsections. Also, we have a new subsection (4.4) that talks on how to assemble the MCF, which includes a new figure (Figure 14) to help readers visualize the final model. 

5) Performance validation parameter and its mathematical derivation could be included. Since the final results could be evaluated some parameters.

Response: Thank you for this comment, however this one we are afraid we cannot completely satisfy right now. Our model is theoretical with a real-world test with empirical parameters. We are providing the initial model and at this moment, we do not perform many theoretical mathematical validations (We do some experimental validations, as well as cost analysis). We stated in the conclusion that our goal was to have it open to the community as quick as possible. We agree that it is important to mathematically verify the parameters and deviations, however if we waited to do so, we would lose timing. We are expecting this framework to be adopted by the community and consequently, the number of experiments and results are added to ours. We are very sorry that at this time, we do not have any mathematical formulation, other than empirical.

6)Language correction must be done as well as grammatical errors can be avoided.

Response: Thank you, we used the MDPI editing service to have our english mistakes corrected by professionals.
